# Fast Stochastic Composite Minimization and an Accelerated Frank-Wolfe Algorithm under Parallelization

**Benjamin Dubois-Taine**
DI ENS, Ecole normale supérieure,
Université PSL, CNRS, INRIA
75005 Paris, France
benjamin.paul-dubois-taine@inria.fr

**Francis Bach**
DI ENS, Ecole normale supérieure,
Université PSL, CNRS, INRIA
75005 Paris, France
francis.bach@inria.fr

**Quentin Berthet**
Google Research, Brain team, Paris
qberthet@google.com

**Adrien Taylor**
DI ENS, Ecole normale supérieure,
Université PSL, CNRS, INRIA
75005 Paris, France
adrien.taylor@inria.fr

## Abstract

We consider the problem of minimizing the sum of two convex functions. One of those functions has Lipschitz-continuous gradients, and can be accessed via stochastic oracles, whereas the other is "simple". We provide a Bregman-type algorithm with accelerated convergence in function values to a ball containing the minimum. The radius of this ball depends on problem-dependent constants, including the variance of the stochastic oracle. We further show that this algorithmic setup naturally leads to a variant of Frank-Wolfe achieving acceleration under parallelization. More precisely, when minimizing a smooth convex function on a bounded domain, we show that one can achieve an $\epsilon$ primal-dual gap (in expectation) in $\tilde{O}(1/\sqrt{\epsilon})$ iterations, by only accessing gradients of the original function and a linear maximization oracle with $O(1/\sqrt{\epsilon})$ computing units in parallel. We illustrate this fast convergence on synthetic numerical experiments.

## 1 Introduction

We consider the composite minimization problem

$$\min_{y \in \mathbf{V}} \left\{ F(y) := G(y) + H(y) \right\}, \tag{1}$$

where $\mathbf{V}$ is some real vector space equipped with a norm $\|\cdot\|$ and an inner product $\langle \cdot, \cdot \rangle$, see Section 2 for details. For instance, one can consider $\mathbf{V} = \mathbb{R}^d$ or $\mathbf{V} = \mathbb{R}^{p \times q}$ equipped with the standard Euclidean inner product and norm. The function $G$ is convex and $\beta$-smooth and the function $H$ is $\mu$-strongly convex, both with respect to (w.r.t.) some norm $\|.\|$ on $\mathbf{V}$ (see Section 2 for precise definitions). We assume that $H$ is "simple", meaning that one can efficiently solve

$$\arg\min_{y \in \mathbf{V}} \left\{ \langle z, y \rangle + H(y) + \alpha D_w(y, z_0) \right\}, \tag{2}$$

where $D_w$ is the Bregman divergence induced by a strongly convex function $w$. For instance, when $w$ is the standard squared Euclidean norm, this amounts to computing the proximal operator of $H$.

36th Conference on Neural Information Processing Systems (NeurIPS 2022).

When deterministic gradients of $G$ are available, an accelerated method relying on (2) and achieving rates of the form $O\left(\exp\left(-k\sqrt{\frac{\mu}{\beta}}\right)\right)$ was proposed by Diakonikolas and Guzmán [13, AGD+].

In this work, we are interested in the case where $G$ is only available through a stochastic oracle. In particular, we provide an accelerated algorithm converging in function values to a neighborhood of the minimum with the same rate as above. The size of the neighborhood is of order $O\left(\frac{\sigma^2}{\sqrt{\mu\beta}}\right)$ where $\sigma^2$ is the variance of the stochastic gradients. The dependence of the noise term on $1/\sqrt{\mu\beta}$ is similar to that of previous stochastic accelerated methods in simpler settings, see, e.g., [3]. Although our rate is an extension of the result in [13], the parameters are different and tailored for the stochastic setup.

In Section 4, we focus on minimization problems over a compact convex set $K$ for which we have access to a linear optimization oracle, just as in the Frank-Wolfe algorithm. Formally, we consider solving $\min_{x\in K} f(x)$ where $f$ is convex with Lipschitz-continuous gradients. In that case, the dual problem can be seen as a particular instance of (1) where $H := f^*$ is the Fenchel conjugate of $f$, and $G$ is a smoothed version of the support function of $K$. Such smoothed functions can only be accessed through a stochastic oracle whose computation boils down to solving a linear optimization problem on $K$. An appropriate choice of $w$ allows us to use the algorithmic framework by only computing gradients of $f$ and solving linear optimization subproblems over $K$. In short, minimizing $f$ over the set $K$ with an $\epsilon$ primal-dual gap requires $\tilde{O}\left(\max\left\{\frac{1}{\sqrt{\epsilon}}, \frac{1}{\epsilon m}\right\}\right)$ iterations when $m$ computing units can be used in parallel (each of them only solving linear optimization subproblems on $K$). In particular, when $m = 1/\sqrt{\epsilon}$, this effectively yields an accelerated rate of $\tilde{O}\left(1/\sqrt{\epsilon}\right)$ iterations with a single gradient evaluation per iteration to achieve the required accuracy.

We emphasize that the first contribution of this work is to provide an accelerated method for general stochastic composite problems (in Section 3). In Section 4, we then show that it can be used for obtaining an accelerated Frank-Wolfe algorithm.

## 1.1 Related Work

As previously emphasized, the main algorithmic ingredients which inspired this work were developed by [13] and [19]. For further references on acceleration and gradient-based composite convex minimization, we refer to the original works [38, 40] or to a recent survey [15]. For the stochastic setup, we refer to [32]. We refer to [39, 12] for further references and pointers on those topics.

Let us briefly describe the main differences between this work and [13, 19]. First, Gasnikov and Nesterov [19] obtain accelerated rates similar to ours when the strong convexity assumption is placed on $G$ instead of $H$ here (which has practical consequences, including well-posedness of (2) and the size of the constants $\beta$ and $\mu$, see discussion in [13, Introduction]). When the gradients of $G$ are stochastic, [19] provides convergence rates, but only when the underlying norm is the Euclidean norm. Finally, whereas Diakonikolas and Guzmán [13] consider more general assumptions in the deterministic non-Euclidean case, our work also covers stochastic approximations. We emphasize that each assumption we make (namely accelerated, stochastic, proximal Bregman methods) is necessary to yield acceleration of Frank-Wolfe under parallelization in Section 4.

Two key observations underlying this work are on the one hand the duality link between Frank-Wolfe methods and Bregman methods [4], and on the other hand randomized smoothing techniques [41, 14, 1, 2, 7, 8] which can be naturally computed using linear optimization steps. The main question of interest was whether accelerated Bregman methods should naturally give rise to accelerated Frank-Wolfe methods on the dual.

The Frank-Wolfe (FW) method (a.k.a. conditional gradient method) and its variants were first introduced by Frank and Wolfe [16] (see also [27, 26] for more modern presentations). When considering optimization over a convex set $K$, classical first-order methods are often naturally embedded with a projection operator (onto $K$). Depending on $K$, those projections are potentially costly. An alternate approach for taking constraints into account within first-order method consists in using linear optimization oracles (a.k.a. Frank-Wolfe techniques). In many applications, such linear minimizations are much cheaper than projecting onto the feasible set, see for example [17, 23]. Despite its wide use in practical applications, the main drawback of the Frank-Wolfe method lies in its slow convergence rate of $O(1/\epsilon)$, standing in sharp contrast with the $O(1/\sqrt{\epsilon})$ convergence of Nesterov's accelerated gradient descent [38] relying on projections.

As "purely accelerated" rates of convergence are out of reach for vanilla Frank-Wolfe methods (see lower complexity bound in [33]), most works on accelerating Frank-Wolfe have focused on exploiting specific additional assumptions on the problems at hand. Common such assumptions include strong convexity of the feasible set [11, 35, 18], and strong convexity of the objective function along with the assumption that the minimizer lies in the interior of the feasible set [21, 31]. In both cases, Frank-Wolfe is known to improve on the $O(1/\epsilon)$ rate. Some efforts have also gone into finding rates matching performances of Nesterov's accelerated gradient descent [38] without strong convexity. In particular, when $K$ is a polytope and when a certain type of constraint qualification is satisfied, Frank-Wolfe converges asymptotically at rate $O(1/k^2)$ [5]. Adding momentum to Frank-Wolfe has also been studied, and accelerated rates are attained on some $\ell_p$-norm balls when the minimizer is on the boundary of the feasible set [36]. Our approach in this work is orthogonal to the aforementioned results, in that we do not make additional assumptions on the objective function or the feasible set, but instead show that parallelization can help reaching accelerated rates for a variant of Frank-Wolfe. Note that [34] manages to reach a similar rate of $O(1/\sqrt{\epsilon})$ iterations for a variant of Frank-Wolfe, where each iteration requires one gradient evaluation and $O(1/\sqrt{\epsilon})$ calls to a linear optimization oracle. However, in contrast with our algorithm, their approach is non-parallelizable as the calls to the linear optimization oracle need to be made sequentially.

The rest of the paper is organized as follows. In Section 2 we define notations and review some classical definitions from convex analysis. In Section 3 we provide worst-case rates for a stochastic Bregman dual-averaging-type algorithm, along with some intuitions and proof sketches. In Section 4 we show how a Frank-Wolfe method directly fits within this framework, thereby obtaining accelerated worst-case guarantees on the primal-dual gap. In Section 5, we illustrate our theoretical findings on a set of simple numerical examples.

## 2 Notation and definitions

Formally, we consider a real finite-dimensional vector space $\mathbf{V}$ and its dual space $\mathbf{V}^*$ consisting of all linear functions on $\mathbf{V}$, as well as a dual pairing denoted by $\langle \cdot, \cdot \rangle : \mathbf{V}^* \times \mathbf{V} \to \mathbb{R}$, and a norm $\|\cdot\| : \mathbf{V} \to \mathbb{R}$. We also consider the corresponding dual norm $\|\cdot\|_* : \mathbf{V}^* \to \mathbb{R}$ induced by the choice of $\|\cdot\|$ and $\langle \cdot, \cdot \rangle$ using $\|z\|_* = \sup_{\|x\| \leq 1} \langle z, x \rangle$. For instance, one can use $\mathbf{V} = \mathbf{V}^* = \mathbb{R}^d$ or $\mathbf{V} = \mathbf{V}^* = \mathbb{R}^{p \times q}$ equipped with the standard Euclidean inner product and norm. We insist on the fact that the formal notation $\mathbf{V}$ and $\mathbf{V}^*$ is used mostly for emphasizing differences between primal and dual spaces/problems below. For a closed, proper, convex function $\Psi : \mathbf{V} \to \mathbb{R}$, we denote $\partial \Psi(x)$ the set of all subgradients of $\Psi$ at $x$. When $\partial \Psi(x)$ is a singleton, we denote its single element by $\nabla \Psi(x)$.

**Definition 1.** $\Psi : \mathbf{V} \to \mathbb{R}$ is L-smooth w.r.t. $\|\cdot\|$ if for all $x, y \in \mathbf{V}$,

$$\|\nabla \Psi(x) - \nabla \Psi(y)\|_* \leq L \|x - y\|. \tag{3}$$

The following proposition will be helpful throughout the paper (see [39] for a proof).

**Proposition 1.** $\Psi : \mathbf{V} \to \mathbb{R}$ is L-smooth w.r.t. $\|\cdot\|$ if and only if for all $x, y \in \mathbf{V}$,

$$\Psi(x) \leq \Psi(y) + \langle \nabla \Psi(y), x - y \rangle + \frac{L}{2} \|x - y\|^2.$$

**Definition 2.** $\Psi : \mathbf{V} \to \mathbb{R}$ is $\mu$-strongly convex w.r.t. $\|\cdot\|$ if for all $x, y \in \mathbf{V}$ and any $g_\Psi(y) \in \partial \Psi(y)$,

$$\Psi(x) \geq \Psi(y) + \langle g_\Psi(y), x - y \rangle + \frac{\mu}{2} \|x - y\|^2. \tag{4}$$

**Definition 3.** For a closed, proper, convex function $\Psi : \mathbf{V} \to \mathbb{R}$, its conjugate function is defined as

$$\Psi^*(z) = \sup_{x \in \mathbf{V}} \langle z, x \rangle - \Psi(x). \tag{5}$$

For later references, we need a few results related to smoothing the support function of a set $K \subset \mathbf{V}^*$ ($K$ will appear as a convex set in the dual problem to (1)).

**Definition 4.** For a set $K \subset \mathbf{V}^*$, we define $I_K$ as the indicator function of $K$, i.e., $I_K(x) = 0$ for $x \in K$ and $I_K(x) = \infty$ for $x \in \mathbf{V}^* \setminus K$. The support function of $K$ is defined as

$$s(y) = \sup_{x \in K} \langle x, y \rangle. \tag{6}$$

If $K$ is non-empty, convex and closed, the indicator and support functions of $K$ are conjugates of each other, i.e., $(I_K)^* = s$ and $s^* = (I_K)^{**} = I_K$ [42]. The support function is convex yet not differentiable in general applications. For the purpose of our work, we consider smoothing the support function via randomization. Such stochastic smoothing has recently gained popularity in the optimization literature, see for example [41, 14, 1, 2, 7]. We define the main tools and state the relevant properties behind this idea.

**Definition 5.** *For a set $K \subset \mathbf{V}^*$, a scalar $\alpha > 0$ and a random variable $\Delta$, we define the smoothed support function of $K$ as*

$$s_\alpha(y) = \mathbb{E}_\Delta \left[ s(y + \alpha\Delta) \right] = \mathbb{E}_\Delta \left[ \sup_{x \in K} \langle x, \, y + \alpha\Delta \rangle \right]. \tag{7}$$

We will use the following proposition on the smoothed support function, proved in [7].

**Proposition 2.** *Suppose $K \subset \mathbf{V}^*$ is convex compact and let $R_K = \max_{x \in K} \|x\|_*$. Suppose the random variable $\Delta$ has positive differentiable density $d\pi(z) \propto \exp(-\eta(z))dz$ for some function $\eta(\cdot)$, and let $M^2 = \mathbb{E}\left[\|\nabla_z \eta(Z)\|_*^2\right]$. Then $s_\alpha$ is convex, $\frac{R_K M}{\alpha}$-smooth w.r.t. $\|\cdot\|$ and for any $y \in \mathbf{V}$,*

$$\nabla s_\alpha(y) = \mathbb{E}_\Delta \left[ \arg\max_{x \in K} \langle x, \, y + \alpha\Delta \rangle \right] \quad \text{and} \quad s(y) \leq s_\alpha(y) \leq s(y) + \alpha s_1(0).$$

## 3 Fast Stochastic Composite Minimization

In this section we focus on solving the composite problem

$$\min_{y \in \mathbf{V}} \left\{ F(y) := G(y) + H(y) \right\}, \tag{8}$$

where $G$ is convex and $\beta$-smooth, and $H$ is $\mu$-strongly convex w.r.t. $\|\cdot\|$. Algorithm 1 summarizes our proposed algorithm, which resembles the general AGD+ algorithm from [13]. Notable differences include access to gradients of $G$ through a stochastic oracle, and explicit dependencies on the smoothness and strong convexity constants for the different updates. Moreover, the presented analysis is different and specifically tailored to handle stochasticity in the gradients of $G$.

An actual implementation of Algorithm 1 requires the ability to efficiently solve the minimization step (13), which should be well-defined. This intermediate minimization subproblem is often referred to as a Bregman proximal problem, and a sufficient condition for this operation to be well-defined is to require $w$ to be strongly convex. In the case where $w$ is the Euclidean norm, (13) amounts to computing the proximal operator of $H$. Considering a general regularizer $w$ has several benefits, in that the Euclidean norm might not well capture the geometry of the problem, and because particular choices of $w$ might make (13) easily solvable. The latter will become particularly clear in Section 4.

Before we move on to the analysis, let us emphasize that the first step of Algorithm 1 consisting in picking $z_0 \in \arg\min_y w(y)$ is not restrictive. Indeed one could instead pick any $z_0 \in \mathbf{V}$ and set $\tilde{w}(y) = w(y) - \langle g_w(z_0), \, y \rangle$ where $g_w(z_0) \in \partial w(z_0)$. One could then run Algorithm 1 with the "shifted" $\tilde{w}$ instead of $w$. Doing so does not change the complexity of the minimization step (13), and the following analysis remains unchanged.

We are now ready to analyse Algorithm 1. For this purpose let us define

$$c_k := \sum_{i=0}^{k-1} (A_{i+1} - A_i)(G(y_i) - \langle \nabla G(y_i), \, y_i \rangle).$$

where the sequences $\{A_k\}_{k \in \mathbb{N}}$ and $\{y_k\}_{k \in \mathbb{N}}$ are defined in Algorithm 1. We also use the notation $\mathbb{E}_k$ for denoting the expectation at iteration $k$ conditioned on the previous iterations (that is, $\mathbb{E}_k$ shortens $\mathbb{E}_k[\cdot] = \mathbb{E}[\cdot \mid y_k, z_k, d_k, c_k]$), while $\mathbb{E}$ denotes the total expectation. Before we can state our results, we need one more assumption on the stochastic gradients.

**Assumption 1.** *For any $k \in \mathbb{N}$, the stochastic gradients satisfy $\mathbb{E}_k[g_k] = \nabla G(v_k)$ and $\mathbb{E}_k \|g_k - \nabla G(v_k)\|_*^2 \leq \sigma^2$.*

The assumption on the variance of the stochastic gradients is common when studying stochastic first-order methods, and allows proving the next proposition, which relates consecutive iterations.

---

**Algorithm 1** Stochastic Composite Minimization

---

**Input**: $(\beta, \mu, \nu)$. $\beta$-smooth function $G$, $\mu$-strongly convex function $H$, $\nu$-strongly convex function $w$, all w.r.t. the same norm $\|\cdot\|$.

Pick $z_0 \in \arg\min_y w(y)$ and set $y_0 = z_0$.

Set $A_0 = 0$ and $d_0 = 0$.

**for** $k = 0, 1, \ldots$ **do**

$$A_{k+1} = \frac{A_k(\mu + 2\beta + \sqrt{\mu\beta}) + \beta\nu + \sqrt{(\beta\nu + \mu A_k)^2 + 4A_k\beta^2\nu + 5A_k^2\mu\beta + 2A_k\sqrt{\mu\beta}(\beta\nu + A_k\mu)}}{2(\beta + \sqrt{\mu\beta})} \quad (9)$$

$$\tau_k = 1 - \frac{A_k}{A_{k+1}} \quad (10)$$

$$v_k = (1 - \tau_k)y_k + \tau_k z_k \quad (11)$$

Compute a stochastic gradient $g_k$ of function $G$ at iterate $v_k$,

$$d_{k+1} = d_k + (A_{k+1} - A_k)g_k \quad (12)$$

$$z_{k+1} \in \arg\min_{y \in \mathbf{V}} \{\langle d_{k+1}, y \rangle + A_{k+1}H(y) + \beta w(y)\} \quad (13)$$

$$y_{k+1} = (1 - \tau_k)y_k + \tau_k z_{k+1} \quad (14)$$

**end**

---

**Proposition 3.** *Suppose Assumption 1 holds and let $m_k(y) := \langle d_k, y \rangle + c_k + A_k H(y) + \beta w(y)$. At iteration $k$, the iterates of Algorithm 1 satisfy*

$$\mathbb{E}_k\left[A_{k+1}F(y_{k+1}) - m_{k+1}(z_{k+1})\right] \leq A_k F(y_k) - m_k(z_k) + (A_{k+1} - A_k)\frac{\sigma^2}{2\sqrt{\mu\beta}}. \quad (15)$$

We defer the full proof to Appendix A, but highlight the main steps here. This way we also hope to shed light on the update of $A_{k+1}$. The first step is to compute an upper bound on $A_{k+1}G(y_{k+1})$ depending on $A_k G(y_k)$. The second step is similar, and computes an upper bound on $A_{k+1}H(y_{k+1})$ depending on $A_k H(y_k)$. Summing up the two inequalities and taking expectations yields inequality (15) with an additional term in $\|z_{k+1} - z_k\|^2$. To exactly obtain inequality (15) we set $A_{k+1}$ so as to cancel out the coefficient multiplying $\|z_{k+1} - z_k\|^2$. This turns out to be equivalent to setting $A_{k+1}$ as the root of a quadratic polynomial, explaining the form of update (9) in Algorithm 1.

Proposition 3 allows us to get the final rate of convergence of Algorithm 1. We again defer the proof to Appendix A. The idea here is first to unroll the recursion in (15) so as to get a constant term on the right-hand side of the inequality. We then relate $m_N(z_N)$ to $m_N(y_\star)$ and to the minimum of $F$ and finally, we show that $A_{k+1} \geq A_k\left(1 + \frac{\sqrt{\mu}}{2(\sqrt{\beta} + \sqrt{\mu})}\right)$, which gives the exponential decay term.

**Theorem 1.** *Suppose Assumption 1 holds, let $y_\star \in \arg\min F$ and define $D_w(y_\star, y_0) = w(y_\star) - w(y_0) \geq 0$. The convergence rate of Algorithm 1 after $k$ iterations is*

$$\mathbb{E}[F(y_k) - F(y_\star)] \leq \exp\left(-k\frac{\sqrt{\mu}}{2\left(\sqrt{\beta} + \sqrt{\mu}\right)}\right)\beta D_w(y_\star, y_0) + \frac{\sigma^2}{2\sqrt{\mu\beta}}. \quad (16)$$

This rate is a typical accelerated rate. In the (usual) case where $\mu \leq \beta$, the exponential decay is bounded above by $\exp\left(-\frac{k}{4}\sqrt{\frac{\mu}{\beta}}\right)$, which shows the natural dependence on $\sqrt{\frac{\mu}{\beta}}$. In addition, the neighborhood term is of the form $O\left(\frac{\sigma^2}{\sqrt{\mu\beta}}\right)$, which is again typical of accelerated stochastic methods [3]. Note that $w(y_\star) - w(y_0)$ is equal to the Bregman divergence $D_w(y_\star, y_0) := w(y_\star) - w(y_0) - \langle g_w(y_0); y_\star - y_0 \rangle$ (with $g_w(y_0) \in \partial w(y_0)$) through the choice $g_w(y_0) = 0 \in \partial w(y_0)$, which is valid as $y_0$ minimizes $w(\cdot)$. We emphasize that we do not require differentiability of the function $w$ anywhere.

In the next section, we show how the above algorithm can be directly applied to a smoothed dual of a minimization problem over a compact convex set, yielding a variant of the Frank-Wolfe algorithm which can achieve accelerated rates under parallelization.

# 4 Accelerating Frank-Wolfe with parallelization

We consider the following minimization problem over a compact convex set $K \subset \mathbf{V}^*$

$$\min_{x \in \mathbf{V}^*} \{ f(x) + I_K(x) \}, \tag{17}$$

where $f$ is convex and $L$-smooth w.r.t. $\|\cdot\|_*$. Its Fenchel-Rockafellar dual [42] reads

$$\max_{y \in \mathbf{V}} \{ d(y) := -s(-y) - f^*(y) \}. \tag{18}$$

The smoothness of $f$ implies that $f^*$ is $1/L$-strongly convex w.r.t. $\|\cdot\|$ [43, Proposition 12.60]. The term in $s$ is however not smooth w.r.t. $\|\cdot\|$, which prevents us from directly applying results from the previous section. Instead, we choose some smoothing parameter $\alpha > 0$ and, following Definition 5 and Proposition 2, we consider the smoothed minimization problem

$$\min_{y \in \mathbf{V}} \{ s_\alpha(-y) + f^*(y) \}. \tag{19}$$

Problem (19) fits within the framework of (8) with $G(y) = s_\alpha(-y)$, $H(y) = f^*(y)$, $\mu = 1/L$ and $\beta = \frac{R_K M}{\alpha}$. Using Proposition 2, we see that an unbiased stochastic gradient $g$ of $G$ at some point $y$ can be obtained by sampling $\Delta$ and computing $g = -\arg\max_{u \in K} \langle u, -y + \alpha\Delta \rangle$. This boils down to a linear optimization oracle over $K$, the same oracle as that of the Frank-Wolfe algorithm.

We now show how to pick the distance generating function $w$ such that the minimization step of Algorithm 1 has a closed form. Recalling that $z_0$ must minimize $w$, we set $w(y) = f^*(y) - \langle g_{f^*}(z_0), y \rangle$ where $g_{f^*}(z_0) \in \partial f^*(z_0)$. Clearly, $w$ is $1/L$-strongly convex w.r.t. $\|\cdot\|$ and is minimized at $z_0$. Moreover, first-order optimality conditions for the minimization step (13) of Algorithm 1 read

$$0 \in d_{k+1} + A_{k+1} \partial f^*(z_{k+1}) + \frac{R_K M}{\alpha} \left( \partial f^*(z_{k+1}) - g_{f^*}(z_0) \right) \tag{20}$$

$$\iff \partial f^*(z_{k+1}) \ni \frac{\frac{R_K M}{\alpha}}{A_{k+1} + \frac{R_K M}{\alpha}} g_{f^*}(z_0) - \frac{d_{k+1}}{A_{k+1} + \frac{R_K M}{\alpha}}. \tag{21}$$

Choosing some $x_0 \in K$ and setting $z_k = \nabla f(x_k)$ for all $k$, one can replace (21) by

$$x_{k+1} = \frac{\frac{R_K M}{\alpha}}{A_{k+1} + \frac{R_K M}{\alpha}} x_0 - \frac{d_{k+1}}{A_{k+1} + \frac{R_K M}{\alpha}}. \tag{22}$$

Doing this allows to avoid computing subgradients of $f^*$. Instead, whenever the value of $z_k$ is needed, we compute $\nabla f(x_k)$. We summarize this fully primal algorithm in Algorithm 2. Note that similar tricks to obtain "dual-free" methods have already been used, see for example [32, 46, 25, 10, 28].

Observe that each iteration of Algorithm 2 requires the computation of one gradient of $f$, and $m$ calls to a linear maximization oracle. When $m = 1$, an iteration of Algorithm 2 is as costly as one iteration of Frank-Wolfe. We now show how to further exploit parallelization. To do so, we need to ensure that using multiple samples appropriately improves the quality of the approximation of the true gradient. This the point of the following assumption.

**Assumption 2.** *There exists a norm-dependent constant $\rho_{\|\cdot\|_*}$ such that the variance verifies*

$$\sigma^2 = \mathbb{E}_k \left\| \frac{1}{m} \sum_{i=1}^{m} g_{k,i} - \nabla G(v_k) \right\|_*^2 \leq \frac{4 R_K^2 \rho_{\|\cdot\|_*}}{m}. \tag{23}$$

**Remark 1.** *A few examples are in order. For the standard Euclidean norm $\|\cdot\| = \|\cdot\|_* = \|\cdot\|_2$ it holds that $\rho_{\|\cdot\|_2} = 1$. For $\ell_p$-norms on a space of dimension $d$, we have*

$$\rho_{\|\cdot\|_p} = \begin{cases} d^{2/p - 1} & \text{for } 1 \leq p < 2 \\ p - 1 & \text{for } 2 \leq p < \infty \\ e^2 (\log d + 1) & \text{for } p = \infty. \end{cases} \tag{24}$$

*In the general case, as all norms are equivalent in finite dimensional spaces, there exist constants $c, C > 0$ such that $c \|\cdot\|_2 \leq \|\cdot\|_* \leq C \|\cdot\|_2$. Then $\rho_{\|\cdot\|_*} \leq \frac{C^2}{c^2}$. We refer to Appendix B.3 for more details and references.*

---

**Algorithm 2** Parallel Frank-Wolfe (PFW)

---

**Input**: $(L, x_0, R_K, M, m, \alpha, T)$. $L$-smooth convex function $f$ w.r.t. $\left\| \cdot \right\|_*$, $x_0 \in K$, $R_K = \max_{x \in K} \left\| x \right\|_*$, distribution with density $d\pi(z) \propto \exp(-\eta(z))dz$ such that $M^2 = \mathbb{E}_\Delta \left\| \nabla \eta(\Delta) \right\|_*^2$, $m$ computing units in parallel, smoothing parameter $\alpha$, number of iterations $T$.

Set $A_0 = 0$, $d_0 = 0$, $\beta = \frac{R_K M}{\alpha}$, $\mu = \nu = \frac{1}{L}$, $y_0 = \nabla f(x_0)$.

**for** $k = 0, 1, \ldots, T - 1$ **do**

$$A_{k+1} = \frac{A_k(\mu + 2\beta + \sqrt{\mu\beta}) + \beta\mu + \sqrt{\mu^2(\beta + A_k)^2 + 4A_k\beta^2\mu + 5A_k^2\mu\beta + 2A_k\mu\sqrt{\mu\beta}(\beta + A_k)}}{2(\beta + \sqrt{\mu\beta})}$$

$$\tau_k = 1 - \frac{A_k}{A_{k+1}}$$

$$v_k = (1 - \tau_k)y_k + \tau_k \nabla f(x_k)$$

For all $i \in [m]$ in parallel, sample $\Delta_i \sim d\pi$, compute $g_{k,i} = - \arg\max_{u \in K} \langle u, -v_k + \alpha \Delta_i \rangle$

$$g_k = \frac{1}{m} \sum_{i=1}^{m} g_{k,i}$$

$$d_{k+1} = d_k + (A_{k+1} - A_k)g_k$$

$$x_{k+1} = \frac{\beta}{A_{k+1} + \beta}x_0 - \frac{1}{A_{k+1} + \beta}d_{k+1}$$

$$y_{k+1} = (1 - \tau_k)y_k + \tau_k \nabla f(x_{k+1})$$

**end**

---

As mentioned earlier, one can directly obtain convergence of the dual problem (19) by applying Theorem 1. The next theorem states that not only are the iterates $\{x_k\}_{k \in \mathbb{N}}$ always feasible, but also that the previous rate of convergence also holds for the primal-dual gap between the iterates $\{x_k\}_{k \in \mathbb{N}}$ and $\{y_k\}_{k \in \mathbb{N}}$. For simplicity of the exposition we assume that $\frac{R_K M}{\alpha} \geq \frac{1}{L}$. In practice this is not an issue since we will choose small values of $\alpha$.

**Theorem 2.** *Suppose $f$ is convex and $L$-smooth w.r.t. $\left\| \cdot \right\|_*$ and let $x_0 \in K$. Under Algorithm 2, the primal iterates are always feasible, i.e., $x_k \in K$ for all $k \in \mathbb{N}$. Moreover, after $k$ iterations, we have*

$$\mathbb{E}\left[ f(x_k) - d(y_k) \right] \leq \exp\left( -k\frac{\sqrt{\alpha}}{4\sqrt{LR_K M}} \right) \frac{R_K M}{\alpha}\left( f(x_0) - f(x_\star) \right) + \frac{2R_K^2 \rho_{\|\cdot\|_*}}{m}\sqrt{\frac{\alpha L}{R_K M}} + \alpha s_1(0).$$

The above convergence result is obtained by first plugging the smoothness and strong convexity constants of this section within the result of Proposition 3. Then, one has to relate $m_k(z_k)$ to the primal objective $f(x_k)$. The first term in the upper bound then follows by recursion. The second term comes from the variance, which can be bounded by $\frac{4R_K^2 \rho_{\|\cdot\|}}{m}$ when $m$ stochastic gradients are computed in parallel. The last term $\alpha s_1(0)$ is due to the error induced by the smoothing of the dual. The full proof can be found in Appendix B. From Theorem 2 we are able to show that Algorithm 2 achieves acceleration under parallelization.

**Theorem 3.** *Suppose $f$ is convex and $L$-smooth w.r.t. $\left\| \cdot \right\|_*$ and let $x_0 \in K$. Under Algorithm 2 with $\alpha = \min\left\{ \frac{\epsilon}{3s_1(0)}, \frac{M\epsilon^2 m^2}{36LR_K^3 \rho_{\|\cdot\|_*}^2} \right\}$, the number of iterations required to achieve an $\epsilon$ primal-dual gap, i.e., $\mathbb{E}[f(x_k) - d(y_k)] \leq \epsilon$, is*

$$k \geq \frac{4\sqrt{LR_K M}}{\sqrt{\alpha}} \log\left( \frac{3R_K M(f(x_0) - f(x_\star))}{\epsilon\alpha} \right). \tag{25}$$

*The complexity is therefore $\tilde{O}\left( \max\left( \frac{1}{\sqrt{\epsilon}}, \frac{1}{\epsilon m} \right) \right)$.*

The above complexity result shows that (i) when $m = 1/\sqrt{\epsilon}$, we get an accelerated rate $\tilde{O}(1/\sqrt{\epsilon})$, (ii) when $m = 1$, we recover the classical $\tilde{O}(1/\epsilon)$ complexity of Frank-Wolfe (up to logarithmic terms), (iii) there is no theoretical gain in going beyond $m = 1/\sqrt{\epsilon}$ computing units in parallel.

Moreover, while the total number of calls to the linear optimization oracle to reach an $\epsilon$ primal-dual gap is the same as in Frank-Wolfe for any value of $m$, in the case where $m > 1$, the number of required

gradients of $f$ is strictly less than in Frank-Wolfe. This is because in each iteration of Algorithm 2 we compute one gradient of $f$ no matter what the value of $m$ is. In particular, when $m = 1/\sqrt{\epsilon}$, we need only $\tilde{O}(1/\sqrt{\epsilon})$ gradients of $f$ in total, compared to $O(1/\epsilon)$ in Frank-Wolfe.

One can note that the proposed algorithm in not as universal as the classical Frank-Wolfe method, as it requires upper bounds on several problem-specific constants. However, acceleration of Frank-Wolfe has been extensively studied in the literature (see Section 1) and to the best of our knowledge our work is the first to provide accelerated rates (under parallelization) without further assumptions on the constraint set and/or the objective function. We argue next that estimating upper bounds on most parameters does not pose significant challenges. Indeed, as the user is free to pick any distribution to sample $\Delta$ from (up to the assumptions of Proposition 2), one can choose a distribution for which the constant $M$ is easy to compute (we give two examples in Section 5). Moreover, the user typically knows the set over which the optimization is carried over, and from such knowledge an upper bound on the diameter can often be easily computed. Finally, we also need to upper bound the Lipschitz constant $L$ of the gradient. This seems to be more of a limiting factor compared to classical Frank-Wolfe, although an upper bound on $L$ is often (but not always) necessary to implement other accelerated versions of Frank-Wolfe [18, 34] as well. While it might be possible to circumvent this problem using some sort of line search within our method, it is not straightforward as the smoothing of the dual induces stochasticity, and obtaining theoretical rates for line search techniques on stochastic objectives is notoriously tedious [9, 44, 45].

Finally, note that it might seem like a small modification to Algorithm 2 could give a variant of Frank-Wolfe capable of handling stochastic gradients of $f$ (for previous works on stochastic Frank-Wolfe, see [24, 20, 37] and references therein). Indeed, one could see the noise in $\nabla f(x_k)$ in the update of $v_k$ as the random variable $\Delta$ itself and study the resulting algorithm. However, the current analysis would only work if the noise on $\nabla f(x_k)$ does not depend on the current iterate. Moreover, even under this simplifying assumption, the result does not follow immediately as the stochasticity would appear in both the updates of $v_k$ and $y_{k+1}$, in contrast with the current analysis.

## 5   Experiments

We first consider minimizing a quadratic function in $\mathbb{R}^d$ over the simplex

$$\min_{x \in K_1} \left\{ f_1(x) := \frac{1}{2} \|Ax - b\|_2^2 \right\} \quad \text{where} \quad K_1 = \left\{ x \in \mathbb{R}^d \mid x \geq 0, \sum_{i=1}^d x_i = 1 \right\}, \quad (26)$$

where $A \in \mathbb{R}^{n \times d}$ and $b \in \mathbb{R}^n$. We choose $\Delta$ to have the Gumbel distribution [22] with location and scale parameters equal to 0 and 1 respectively. In this case $R_{K_1} = 1$ and $M = \sqrt{d}$ (see Appendix C for details). We set $n = 200$, $d = 50$ and compare the bound from Theorem 2 with the practical performance of Algorithm 2 for both $m = 1$ and $m = 1/\sqrt{\alpha}$ parallel computer(s). Fig. 1 shows that our upper bound captures well the speed of convergence to a neighborhood.

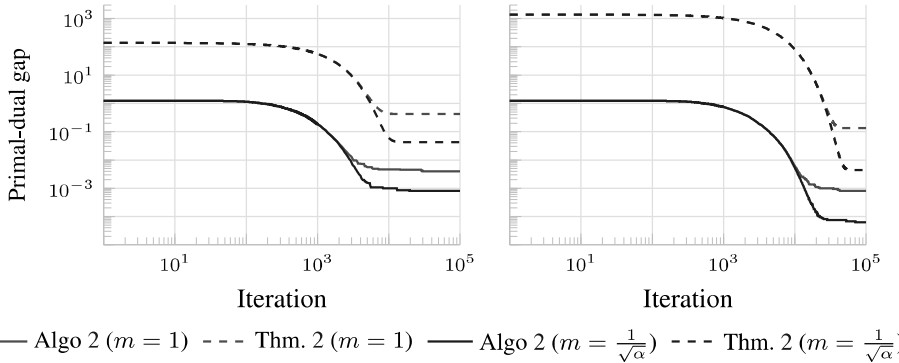

Figure 1: Comparisons between the behavior of Algorithm 2 and that of its theoretical upper bound (see Theorem 2) on a least-squares problem on the simplex for $\alpha = 10^{-2}$ (left) and $\alpha = 10^{-3}$ (right). The plots report the value of the best primal-dual gap incurred at the current iteration.

Next, in order to circumvent the inevitable stalling due to the fixed value of $\alpha$ observed above, we suggest a restarted algorithm which decreases the value of $\alpha$ during training. Starting from $\alpha = 1$ and some $x_0 \in K$, we run Algorithm 2 for $T_\alpha$ iterations with $m_\alpha$ computers in parallel to obtain some approximate solution $x_\alpha \in K$. We then decrease $\alpha$ by a constant factor $c < 1$ and run Algorithm 2 again with the new value of $\alpha$ starting from $x_\alpha$. We repeat this process until a satisfying solution is obtained. We formalize this procedure in Algorithm 3. In practice we choose $c = 0.5$ and set $T_\alpha = \sqrt{\frac{L}{\alpha}} \log \frac{1}{\alpha}$ as a simplified version of the bound (25), and run the above procedure for both $m_\alpha = 1$ and $m_\alpha = \sqrt[1]{\sqrt{\alpha}}$ computers in parallel.

---

**Algorithm 3** Restarted Parallel Frank-Wolfe (R-PFW)

---

**Input**: $(L, x_0, R_K, M, c)$. $L$-smooth convex function $f$, $x_0 \in K$, $R_K = \max_{x \in K} \|x\|_*$, distribution with density $d\pi(z) \propto \exp(-\eta(z))dz$ such that $M^2 = \mathbb{E}_\Delta \|\nabla \eta(\Delta)\|_*^2$, decreasing factor $c$.
Set $\alpha = 1$.
**for** $i = 0, 1, \ldots$ **do**
     Set $T_\alpha = \sqrt{\frac{L}{\alpha}} \log \frac{1}{\alpha}$ and either set $m_\alpha = \frac{1}{\sqrt{\alpha}}$ or $m_\alpha = 1$.
     $x_\alpha = $ Algorithm $2(L, x_0, R_K, M, m_\alpha, \alpha, T_\alpha)$
     Set $x_0 = x_\alpha$ and $\alpha = c\alpha$.
**end**
Return $x_\alpha$.

---

We test the restarted scheme on problem (26) as well as on a generalized matrix completion problem over the trace norm ball,

$$\min_{X \in K_2} \left\{ f_2(x) := \frac{1}{2} \|CX - D\|_F^2 \right\} \quad \text{where} \quad K_2 = \left\{ X \in \mathbb{R}^{p \times q} \mid \|X\|_{tr} \leq 1 \right\}, \quad (27)$$

where $C \in \mathbb{R}^{p \times p}$ and $D \in \mathbb{R}^{p \times q}$. Here $\|\cdot\|_F$ stands for the Frobenius norm and $\|\cdot\|_{tr}$ is the trace, or nuclear, norm. We choose $\Delta$ to have entries distributed according to a standard normal distribution. In this case $R_{K_2} = 1$ and $M = \sqrt{pq}$. We compare the restarted scheme with the Frank-Wolfe algorithm with step-size $\eta_k = \frac{2}{k+1}$ (denoted FW in the plots) and with exact line search (denoted FW-LS in the plots). We set $p = 10$, $q = 8$ and plot the results in Fig. 2. For both problems, we observed that using $M = 1$ instead of the theoretical value ($M = \sqrt{d}$ for (26) and $M = \sqrt{pq}$ for (27)) led to significant speedups. We consequently plot both strategies. We observe that for $m_\alpha = \sqrt[1]{\sqrt{\alpha}}$, Algorithm 3 significantly outperforms Frank-Wolfe, and for $m_\alpha = 1$ the performance is similar to Frank-Wolfe, which was expected from the discussion at the end of Section 4. Full code to reproduce the experiments can be found at the following link: `https://github.com/bpauld/PFW`.

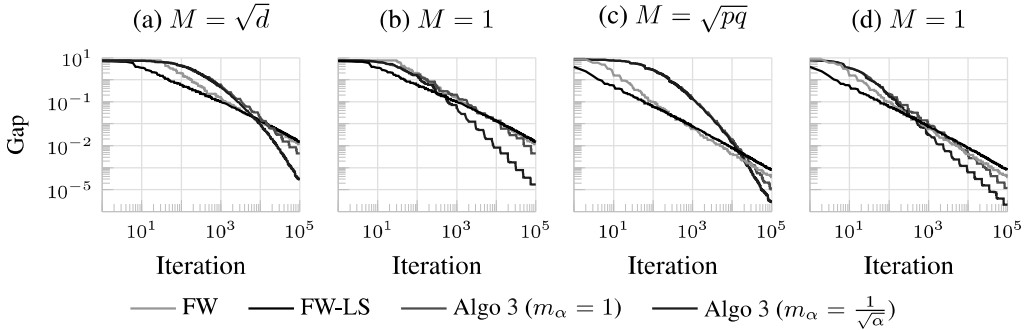

Figure 2: Comparisons between Frank-Wolfe and the restarting scheme (Algorithm 3): a least-squares problem on the simplex ((a) and (b)), and a generalized matrix completion problem on the trace ball ((c) and (d)). The plots report the value of the best primal-dual gap incurred at the current iteration.

# 6 Conclusion

In this work we introduced a stochastic proximal Bregman algorithm able to converge to a neighborhood of the solution at the same accelerated rate as its deterministic counterpart. We then used it to design a variant of the Frank-Wolfe algorithm able to achieve accelerated rates under parallelization. One drawback is that the resulting algorithm is not "any time", in the sense that the desired precision $\epsilon$ must be given as an input to the algorithm. We circumvent this by designing a simple heuristic which slowly decreases the value of $\epsilon$. We plan to investigate the theory for this restarted algorithm in the future. We also hope to use randomized smoothing techniques similar to the ones used in the work to obtain stochastic Frank-Wolfe methods.

# 7 Acknowledgements

The authors acknowledge support from the European Research Council (grant SEQUOIA 724063). This work was funded in part by the french government under management of Agence Nationale de la recherche as part of the "Investissements d'avenir" program, reference ANR-19-P3IA-0001 (PRAIRIE 3IA Institute).

The authors would like to warmly thank Alexander Gasnikov for very constructive remarks and for pointing out the existence and relationship of our work with [34, 20, 28, 10].

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
