# OpenReview forum: "Fast Stochastic Composite Minimization and an Accelerated Frank-Wolfe Algorithm under Parallelization"
_NeurIPS.cc/2022/Conference — NeurIPS 2022 Accept_

### Official Review · Reviewer_BdLs · 2022-07-09

**Rating:** 7
**Confidence:** 3
**Soundness:** 4 excellent
**Presentation:** 3 good
**Contribution:** 3 good

**Summary:**

In this paper, the authors start with a general framework that can solve composite problem G(y)+H(y) where G can only be accessed via a stochastic gradient oracle, with a proximal Bregman-type algorithm. Explicit rate is given along with the final radius of convergence for the framework. This is in turn applied to solving a constrained convex function, where the framework is evoked for its smoothed dual problem. The requirement on stochastic gradient boils down to a linear optimization oracle over the set, which is amenable to parallelization thanks to the smoothing operation. The resulting Frank-Wolfe-type algorithm is shown to have a convergence rate in the duality gap of O(1/sqrt(eps)) with 1/sqrt(eps) parallel queries. Numerical experiments are also conducted on the proposed algorithm.

**Questions:**

- This is probably just confusion on my part. Right before (22), $z_k$ is set to be $\nabla f(x_k)$ for all k, so the resulting Algorithm 2 only involves $x_k,y_k$ and $v_k$. But is the reason I should think of $x_k$ as being the current primal iterate because $y_k^* = \nabla f(x_k^*)$ for the optimal primal-dual pair on (17) and (18), and both $y_k,z_k$ live in the dual space?
- There's another factor of $(1-e^{-z_1})$ missing on the last line of page 24?

**Limitations:**

Yes.

**Strengths And Weaknesses:**

I find the paper generally well-written and think the result is of interest to the community. This reduction from accelerated Bregman method to accelerated Frank-Wolfe algorithm based on parallelization is a nice addition to the literature.

---

> ### Author Response · Authors · 2022-08-01
> **Answer to reviewer BdLs**
>
> We thank the reviewer for their overall positive feedback and review of our work. Their summary also very accurately describes our work; we answer to the two detailed questions below.
>
> 1. It is correct to think of $x_k$ as living in the primal space, while $v_k$, $y_k$ and $z_k$ live in the dual space. Moreover, as the result in Theorem 3 states, $(x_k, y_k)$ can indeed be considered to converge to an optimal primal-dual pair. Regarding the notation $x_k^*$ and $y_k^*$, we do not use it in this work. We hope that we have well understood your question and would be happy to provide further clarification otherwise.
>
> 2. Thank you for the catch, we were indeed missing a $(1 - e^{-z_1})$ factor. We have fixed this in the new version of the paper: It was a typo and did not affect the computation.

---

### Official Review · Reviewer_rQnJ · 2022-07-12

**Rating:** 7
**Confidence:** 3
**Soundness:** 3 good
**Presentation:** 3 good
**Contribution:** 3 good

**Summary:**

In this paper, the authors consider convex minimization problem over bounded domain.
First, they propose a new accelerated gradient method that has several distinguising assumptions and features:
- it works with stochastic gradients of the smooth part;
- it uses strong convexity of the composite part;
- it employs an arbitrary strongly convex prox function as a regularizer.

Then, the authors demonstrate that this method can be applied for solving a smoothed dual problem of miniminizing a smooth function over a compact convex set. In this case, the new method has an interpretation of Frank-Wolfe algorithm. To decrease the variance of the stochastic estimation of the dual gradients, one can use a parallel mini-batching, which provides the method with a provable acceleration given a set of parallel computational units. Numerical experiments on synthetic data are provided.

**Questions:**

Minor questions and remarks:

1. Algorithm 2: this is not clear, what is $du$ from which $\Delta$ is sampled.

2. It might be worth to move the choice of $\alpha$ into the beginning of Theorem 3 (e.g. Let $\alpha = ...$. Now it is easy to miss it).

3. Is it correct that $m = 1$ does not correspond to the 'pure' Frank-Wolfe method? The method seems to be always stochastic.

4. Is it possible to use some line search in experiments?

**Limitations:**

--

**Strengths And Weaknesses:**

The paper is well written. I found the results very interesting and significant. Although, all the building block are quite known in optimization community (accelerated methods, duality, bregman distances, smoothing, etc.), the whole approach fits perfectly together and provides the reader with a number of nice and useful observations.

In particular, it is shown that minibatching for improving stochastic estimation of the gradient of the smoothed support function results in doing in parallel several linear minimization oracle calls for the corresponding convex set. The last operation is the most expensive in the Frank-Wolfe method. Then, the authors demonstrate that choosing the 'batchsize' of size $m$, the resulting complexity is $O( \frac{1}{\sqrt{\epsilon}} + \frac{1}{m \epsilon} )$. Hence, for $m = \frac{1}{\sqrt{\epsilon}}$, this gives the optimal rate. The most imporant feature of this result is that this 'minibatching' is done in parallel, which means that we can get a significant acceleration having several computational units.

In my opinion, there are several limitations of this technique. Probably, it might be benefitial for the presentation to address some of them:

1. The main requirement of the approach is to have an access to a probability distribution with 'positive differentiable density'. It is not thoroughly discussed how difficult is it to find such a distribution, which is suitable for a given problem. Assumption 2 on the variance decrease seems to be also related. It defines a certain constant $\rho$ that depends on the distribution and needs to be known by the method (as for the choice of $\alpha$).

2. Also, the method needs to be given several other paramers (Lipschitz constant of the gradient, diameter of the set, parameter of the distribution $M$). The classical Frank-Wolfe method known to be quite universal: no knowledege of any of this parameters is required. Morevore, it is possible to use a line search in each iteration, that improves convergence significantly. This is not clear, how practical the requirement of knowing all these parameters for the new method, and is it possible to eliminate them.


------------------
After rebuttal:
------------------

I vote for accepting this paper since I believe that the contribution of the paper and the new methods are solid.

---

> ### Author Response · Authors · 2022-08-01
> **Answer to reviewer rQnJ**
>
> We thank the reviewer for their overall positive feedback and review of our work. Their summary is very accurate. We answer their main questions and comments below.
>
>
> **Strength and weaknesses:**
> 1. The requirement that the probability distribution has positive differentiable density is made for technical purposes in order to prove Proposition 2 (which is done in [1]). In practice, note that this requirement is satisfied for many commonly-used distributions (multivariate normal, Gumbel, …), and it is not a difficulty at all.
>
>    On the other hand, Assumption 2 on the variance does not depend on the distribution. Rather, it depends on the norm used to measure the smoothness of the function and on the resulting diameter of the constraint set. In particular, the constant ρ only depends on the underlying norm. In Remark 1, we compute the value of $\rho$ for commonly-used norms. In particular, for the case of the Euclidean norm we have $\rho= 1$, for the $\ell_1$-norm $\rho= d$ and for the $\ell_\infty$-norm $\rho = O(\log d)$.
>
>    As for the choice of $\alpha$, it is tightly related to the choice of $M$, $L$ and $R_K$, which we discuss in the next remark.
>
> 2. We acknowledge that our proposed algorithm is not as universal as the classical Frank-Wolfe method. However, acceleration of Frank-Wolfe methods has been studied in the literature for a long time, and our work is the first to provide accelerated rates (using parallelization) without further assumptions on the constraint set and/or the objective function.
> We now give deeper insights into the choice of the parameters:
>    - Generally speaking, only upper bounds on all parameters are sufficient. In practice, it seems relatively reasonable and this is what we did in our experiments. Let us be a bit more specific and discuss each parameter.
>    - The parameter of the distribution $M$: note that the user of the algorithm is free to choose practically any distribution (up to the assumptions of Proposition 2 discussed above). In particular, the user can always choose a distribution for which $M$ is easy to compute. This is what we do in the experiments, where we give the value of $M$ for the normal and Gumbel distributions.
>    - The diameter of the set: we acknowledge that compared to classical Frank-Wolfe, this is a limitation, although we argue that it is not a major one. Indeed, in most practical applications the user knows the set over which the optimization is carried on. From that knowledge, obtaining an upper bound on the diameter is usually straightforward.
>    - Lipschitz constant of the gradient: we acknowledge that this might be a limitation (if no suitable upper bound can be found), but as we discussed above, the goal of this work is not to be as universal as classical Frank-Wolfe, but rather to fill a missing gap in the literature. Moreover, we point out that several (but not all) other works that attempt to accelerate Frank-Wolfe methods assume knowledge of the Lipschitz constant, see for example [2, 3], and that upper bounds can often be computed in practice.
> Finally, it might be possible to use some type of linesearch within our method. From what we can tell, it is not straightforward as the smoothing of the dual induces stochasticity, and it is known that obtaining theoretical rates for linesearch techniques on stochastic objectives is a tedious task. We believe this is an interesting direction of future research.
>
> [1] Q. Berthet, M. Blondel, O. Teboul, M. Cuturi, J.-P. Vert, and F. Bach. Learning with differentiable perturbed optimizers. 2020.
>
> [2] D. Garber and E. Hazan. Faster rates for the Frank-Wolfe method over strongly-convex sets. 2015.
>
> [3] G. Lan and Y. Zhou, Conditional gradient sliding for convex optimization. 2016
>
> **Questions:**
>
> 1. Thank you for the catch, this should indeed be $d\pi$. We have fixed this in the new version of the paper.
>
> 2. We have made this change in the new version of the paper.
>
> 3. It is true that when $m=1$, our method does not correspond to the pure Frank-Wolfe method, as there is still stochasticity involved. However, when $m=1$, our claim is that the rate of our method matches in expectation the rate of the pure Frank-Wolfe (up to log terms).
>
> 4. Using a linesearch with our method proved to be unsuccessful. As mentioned above, we believe this is due to the induced stochasticity, which tends to make linesearch techniques fail.
>
>    For completeness, we have added to the new version of the paper a comparison with classical Frank-Wolfe with an exact linesearch (for which there is a simple closed-form formula since our experiments deal with quadratics), see Figure 2 of the new version of the paper (see PDF). As the plots show, the exact linesearch does not lead to significant improvements over the simple $\frac{2}{k+1}$ stepsize strategy. This is also in line with the lower complexity bounds for Frank-Wolfe-type methods (see, e.g., [24, Section 3] or [29, Theorem 1]).

---

> > ### Comment · Reviewer_rQnJ · 2022-08-09
> > **re: Answer**
> >
> > Thanks a lot for your answers.
> > Please, consider adding your clarifications to the final version of the paper as they seem to be helpful.

---

### Official Review · Reviewer_3QVT · 2022-07-18

**Rating:** 3
**Confidence:** 4
**Soundness:** 2 fair
**Presentation:** 1 poor
**Contribution:** 1 poor

**Summary:**

The authors propose a Frank-Wolfe-based algorithm for solving partially stochastic composite minimization problems over general convex constraints which admit a regularized linear minimization oracle. This general algorithm is applied to the case where the stochastic part of the objective is a finite sum and propose a parallelized step in which all components of the finite sum are computed in parallel. There are numerical results on least squares with L1-norm constraint and matrix factorization.

**Questions:**

1. There seems to be a mismatch between the experiments and the theory. Is there an experiment that has nonzero H?
2. Is this an actual parallelized algorithm or does it simply have a parallel step (computation of a stochastic gradient)?

**Strengths And Weaknesses:**

1. The paper is difficult to parse. The significance is unclear, especially given what appears to be a significant gap between the theory and the experiments.
2. The presentation of the parallelized algorithm has the flavor of being dishonest since it does appear to be a truly parallel algorithm. If I understand correctly, with the authors view, classical SGD on a finite-sum problem can all be trivially "parallelized."
3. The experiments are very far away from the theory. Interestingly, this would enable the comparison to more recent stochastic FW algorithms rather than vanilla FW.

---

> ### Author Response · Authors · 2022-08-01
> **Answer to reviewer 3QVT**
>
> Thank you for your review of our work. We would like to clarify a few points regarding its objective, the problems that we solve, and how we solve them. Your comments seem to indicate that you have an understanding of our work that is different from the one we were aiming to present. We therefore bring these clarifications before replying to your questions individually.
>
> We have also added further clarification in our main text, hoping that this will avoid some potential confusion for future readers. We thank you for giving us the opportunity to do so.
>
> ### Section 3 (General case)
>
> We tackle the problem of
>
> $ \min_{y \in V} G(y) + H(y) $
>
> - $G$ convex, $\beta$-smooth. → Access to stochastic gradients
> - $H$ $\mu$-strongly convex. → Access to prox operator (see Eq. (2))
>
> We present an algorithm using these information accesses to $G$ and $H$, and give guarantees on its convergence. It does not refer directly to a linear optimization oracle or to Frank-Wolfe.
>
> ### Section 4 (Special case)
>
> We apply this to the special case of
>
> $\min_{x \in V*} f(x) + I_K(x)$ [the primal] ←→ $\min_{y \in V} s_{\alpha}(-y) + f^*(y)$ [the $\alpha$-smoothed dual]
>
> where the dual problem is an instance of the general case studied in Section 3, as suggested by the choice of variable names
>
> - $s_{\alpha}(-y)$ plays the role of $G(y)$. → Access to stochastic gradients through a perturbed linear oracle on $K$, **parallelization here** used to reduce its variance without greater computation time, on algorithm-introduced stochasticity.
> - $f^*$ plays the role of $H$ → A special algorithmic technique allows to only use the gradient of f to solve the prox operator, without computing gradients of $f^*$ (in short, we choose the prox-function $w(\cdot)$ using $f^*$ ).
>
> Note in particular that we consider that we can directly access the gradient of the objective $f$. The only parallel aspect is in reducing the variance of the stochastic gradient of $s_{\alpha}$, the smoothed version of the support function of $K$.
>
> ### Summary
>
> Our objective is general composite optimization, applied in a particular case to the dual of a constrained optimization problem on $K$.
>
> In Section 4, the target optimization problem is not a stochastic one: stochasticity is introduced only in the algorithm for handling the constrained set, and not the objective function itself. We also do not compute all the components of the finite sum in parallel, as the stochasticity does not involve a finite sum. Finally, please note again that stochasticity is used for handling the constraint set, and that our claim related to parallelization is standard.
>
> We hope that this will clarify the aim of our paper, and address some of the points raised in your review, and now reply to your questions individually.
>
> ### About your questions:
>
> (1) We want to highlight again that in Section 4, and in the experiments, the role of $H$ is played by $f^*$ (respectively $f^*_1$ and $f^*_2$ in our experiments). It is never 0. Conversely, $G$ is the smoothed support function of $K$, noted $s_\alpha$ (up to a minus sign). Our experiments are therefore a direct illustration of our theoretical results in 4.
>
> Note that in particular we do not use stochastic gradients of the objective function, so it would not be appropriate to compare with stochastic FW methods. As suggested by another reviewer, we have added a FW with linesearch to the vanilla FW results that we already had.
>
> Regarding the comparison between theoretical bounds and experimental results, their shapes match well : our theoretical analysis accurately predicts the behavior of the optimization error through the steps, and they differ by a multiplicative constant only. This is unavoidable due to the nature of worst-case analysis (which accounts for global problem constants, whereas local constants can only be better).
>
>
> (2) As explained above, we want to clarify that the “parallel” aspect of our algorithm refers to the ability to reduce the variance of gradient estimation for an algorithm-introduced stochastic smoothing, in Section 4. Please note that in this section, we do not parallelize the computation of gradients of the objective function $f$ : the access to the objective function $f$ is assumed deterministic for our Frank-Wolfe algorithm and there is no finite-sum structure assumed. We do parallelize the independent perturbed instances of linear minimization oracle to reduce the variance.

---

### Author Response · Authors · 2022-08-09
**Thank you for the discussion**

We would like to thank the area chair and reviewers for their work on this submission, and for the fruitful discussion that we had with them. As recommended by Reviewer rQnJ, we will incorporate these clarifications to the final version of the paper.

---

### Comment · Area_Chair_L6fZ · 2022-08-09
**Question regarding theoretical novelty**

Dear authors,

I would like to request some clarification about the following question: which (if any) of the complexity bounds in the paper are new?

In particular, it seems that the rate of convergence given in Theorem 1 has already been shown (under the same assumptions*) by Ghadimi and Lan in "Optimal Stochastic Approximation Algorithms for Strongly Convex Stochastic Composite Optimization, II: Shrinking Procedures and Optimal Algorithms."

Moreover, it appears that the algorithm developed in Section 4 can use any method achieving the rate of Theorem 1 as a black box, and consequently the novelty of the bounds of that section is also unclear.

I am sorry that this question comes late in the discussion period, but I thought some opportunity to respond is better than not all. If the option to respond is no longer available by the time you are ready to do so, please consider contacting the program committee for assistance.

\* Ghadimi and Lan write the strong convexity assumption on the smooth component of the objective, but their proof trivially extend to the case that only the sum of the objective and the composite term is strongly convex.

---

> ### Author Response · Authors · 2022-08-10
> **Reply to AC direct question**
>
> Dear AC,
>
> Thank you for your question. We are replying now, as soon as possible after your question, because you asked this directly to us, expecting an answer.
>
> The work that you reference does not put in question our novelty claims, as explained in the following points. Overall, there are indeed many papers on very closely related topics (the closest ones probably being those already mentioned in our work (see [12] Diakonikolas and Guzmán and [18] Gasnikov and Nesterov), but, as discussed in the paper, the ingredients required for our analyses are not present in the works we are aware of, including that of Ghadimi and Lan. In more details:
>
> (1) Ghadimi and Lan consider general norms, but the strong convexity and smoothness are placed on the same function. For general norms, moving strong convexity from one function to another is not an option (contrary to the Euclidean case), so one cannot do so trivially. This essential fact (that of not being able to move strong convexity from one side to the other) motivates a number of works on the topic, including, e.g., [12].
>
>
> (2) More importantly, the convergence rate in Ghadimi and Lan is a convergence to the exact minimizer, with rate of the form $O(\sigma^2/ (\mu \epsilon) + \sqrt( \beta / \epsilon) )$ (using notation from our paper). On the contrary, we prove convergence only up to a ball around the minimizer, but with a much better accelerated linear rate of the form $O( \sqrt( \beta / \mu) \log 1/\epsilon)$.  Note that this accelerated linear rate is absolutely necessary to obtain the result in Section 4 which could not be obtained from the results in Lan, to the best of our knowledge.
>
> As for what concerns the second claim by the AC, it may be true that some other algorithms achieving the same rate as Algorithm 1 of our paper could be used as a black-box to design accelerated Frank Wolfe methods (although to the best of our knowledge, no such algorithm has been studied in the literature). However one needs to be extra careful, as we apply Algorithm 1 to the stochastically smoothed dual of the constrained optimization problem introduced in Section 4. It is thus only by a careful choice of the distance generating function $w$ that we are able to develop an algorithm that does not require computing values or gradients of the conjugate function $f^*$. It is not guaranteed that such a trick would work for some (hypothetical) algorithm achieving the same rate as that of Algorithm 1.
>
> In summary, to the best of our knowledge, the question of accelerating Frank-Wolfe is still open in general, and we are the first to provide an answer under the (stochastic) parallel linear optimization oracle. This was the original intent of our work, and it indeed all boils down to the fact that “reduction from accelerated Bregman method to accelerated Frank-Wolfe algorithm based on parallelization is a nice addition to the literature” (see reviewer BdLs’ comments), and the complexity analyses follow in the same way. From what we can tell, the required accelerated Bregman method has not been studied so far, and we therefore keep all our novelty claims unchanged.
>
> We would be happy to cite this work and to explain these differences if you find it useful, and would be glad to further expand on the topic if need be.
>
> Best,
>
> The authors.

---

### Meta-Review · Area_Chair_L6fZ · 2022-08-26

**Recommendation:** Accept
**Confidence:** Certain

**Metareview:**

The authors design an algorithm for composite stochastic optimization that leverages both smoothness and strong convexity with respect to the same (general) norm, using a stochastic counterpart to recent work by Diakonikolas and Guzman. They then show how to leverage this algorithm and randomized smoothing in order to create an algorithm for constrained smooth convex optimization based on exact gradient evaluations and linear optimization computations. Compared to Frank-Wolfe, the algorithm requires strictly less gradient evaluations and parallelizes the same amount of linear optimization computations.

The paper received generally favorable reviews, with the exception of reviewer 3QVT who did not engage in discussion and whose critique I found unclear. I agree with reviewer rQnJ’s assessment that even though “all the building block are quite known in optimization community (accelerated methods, duality, Bregman distances, smoothing, etc.), the whole approach fits perfectly together and provides the reader with a number of nice and useful observations.” Consequently, I recommend acceptance.

**Award:**

No

---

### Decision · Program_Chairs · 2022-09-14

Accept